# Let the Flows Tell: Solving Graph Combinatorial Optimization Problems with GFlowNets

**Dinghuai Zhang**[*]
Mila

**Hanjun Dai**
Google DeepMind

**Nikolay Malkin, Aaron Courville, Yoshua Bengio, Ling Pan**
Mila

## Abstract

Combinatorial optimization (CO) problems are often NP-hard and thus out of reach for exact algorithms, making them a tempting domain to apply machine learning methods. The highly structured constraints in these problems can hinder either optimization or sampling directly in the solution space. On the other hand, GFlowNets have recently emerged as a powerful machinery to efficiently sample from composite unnormalized densities sequentially and have the potential to amortize such solution-searching processes in CO, as well as generate diverse solution candidates. In this paper, we design Markov decision processes (MDPs) for different combinatorial problems and propose to train conditional GFlowNets to sample from the solution space. Efficient training techniques are also developed to benefit long-range credit assignment. Through extensive experiments on a variety of different CO tasks with synthetic and realistic data, we demonstrate that GFlowNet policies can efficiently find high-quality solutions. Our implementation is open-sourced at https://github.com/zdhNarsil/GFlowNet-CombOpt.

## 1 Introduction

Combinatorial optimization (CO) is a branch of optimization that studies problems of minimizing or maximizing some cost over a finite feasible set. CO problems usually involve discrete structures, such as graphs, networks, and permutations, and require optimizing an objective function subject to discrete constraints in the solution space, which is often NP-hard. CO problems have broad applications, including in medicine, engineering, operations research, and management (Paschos, 2010), and have spurred the development of discrete mathematics and theoretical computer science for a century (Kuhn, 1955; Kruskal, 1956; Ford & Fulkerson, 1956).

During the past few decades, researchers have developed numerical solvers such as GUROBI (Gurobi Optimization, 2023) to give approximate solutions via integer programming. In recent years, interest in

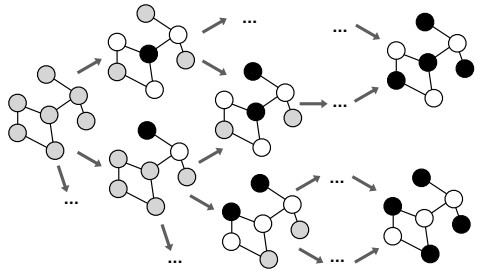

Figure 1: Illustration of GFlowNet for a toy MIS problem where all the states form a DAG. Every trajectory starts from the same initial state (whose vertices are all gray). Each transition denotes adding one vertex to the solution set, *i.e.*, changing one vertex to black. See Section 3.2 for details.

learning-based methods for solving CO problems has grown significantly. These approaches leverage the power of deep networks to learn the inherent structure of CO problems and provide efficient

---

[*]Correspondence to `dinghuai.zhang@mila.quebec`.

37th Conference on Neural Information Processing Systems (NeurIPS 2023).

and effective solutions. One family of machine learning methods utilizes solver-found solutions to provide a supervised learning (SL) signal for training neural networks (Selsam et al., 2018; Gasse et al., 2019; Nair et al., 2020). This line of algorithms requires expensive precomputation by the numerical solver to produce supervision labels. On the other hand, unsupervised learning (UL) methods search for the solution without the help of a solution oracle. One branch of UL methods, called probabilistic methods, decodes the heatmap generated from one pass of a neural network to get solutions (Toenshoff et al., 2019; Karalias & Loukas, 2020). These methods can achieve fast inference at the cost of a large optimality gap. Another branch of unsupervised methods is reinforcement learning (RL), which iteratively refines or constructs the problem solution with practitioner-specified MDPs (Bello et al., 2016; Deudon et al., 2018; Wu et al., 2019).

Despite many recent efforts to apply deep RL to CO problems, such approaches have fundamental limitations. For example, due to the symmetry in problem configurations, there could be multiple optimal solutions to the same CO problem (Li et al., 2018). Standard RL algorithms such as Fujimoto et al. (2018) are grounded in the nature of cumulative reward maximization and fail to promote diversity in the solutions. Although entropy-regularized RL (Haarnoja et al., 2017, 2018; Zhang et al., 2023b) converges to a stochastic policy instead of a deterministic one, these methods target *trajectory-level entropy* rather than *solution-level entropy*. Consequently, the agent may get trapped in solutions that can be reached by many trajectories and lack the ability to generate diverse candidate solutions. In addition, the performance of RL methods largely depends on the designed reward function and relies on a dense per-step reward for learning the value functions and policies. As a result, it is challenging to apply RL in our problems if only a terminal reward is provided.

Although there are attempts to fix these issues (Kwon et al., 2020; Ahn et al., 2020), they are mostly problem-specific and only achieve marginal improvement. In this work, we turn to a more principled framework, namely generative flow networks (Bengio et al., 2021, GFlowNets), to search for high-quality diverse candidates in CO problems. GFlowNet is a novel decision-making framework for learning stochastic policies to sample composite objects with probability proportional to a given terminal reward, which is suitable for problems where the solution is only related to the terminal state of generative trajectories. We design MDPs for a variety of NP-hard CO problems, where the intermediate states form a flow network in the latent space, and GFlowNet learns an agent to sequentially make decisions in this environment (Figure 1). Another challenge of applying GFlowNets is learning from long trajectories. In large-scale graph CO problems, the GFlowNet agent will encounter a very long trajectory before termination, rendering the task of credit assignment challenging. To this end, we develop efficient learning algorithms to train GFlowNets from transitions rather than complete trajectories, which greatly helps the learning process, especially in large-scale setups. Through extensive experiments on different CO tasks, we demonstrate the advantage of our proposed GFlowNet approach. In summary, our contributions are as follows:

- We design a problem-specific MDP for GFlowNet training on four different CO tasks.

- We propose an efficient GFlowNet learning algorithm to enable fast credit assignment for the GFlowNet agents with long trajectories that emerge in our graph CO problems.

- The empirical advantage of GFlowNets is validated through experiments on different CO problems.

## 2 Preliminaries

### 2.1 GFlowNets

Generative flow networks, or GFlowNets, are variational inference algorithms that treat sampling from a target probability distribution as a sequential decision-making process (Bengio et al., 2021, 2023). We briefly summarize the formulation and the main training algorithms for GFlowNets.

We assume that a fully observed, deterministic MDP with set of states $\mathcal{S}$ and set of actions $\mathcal{A} \subseteq \mathcal{S} \times \mathcal{S}$ is given. The MDP has a designated *initial state*, denoted $\mathbf{s}_0$. Certain states are designated as *terminal* and have no outgoing actions; the set of terminal states is denoted $\mathcal{X}$. All states in $\mathcal{S}$ are assumed to be reachable from $\mathbf{s}_0$ by a (not necessarily unique) sequence of actions (see Figure 2). A *complete trajectory* is a sequence of states $\tau = (\mathbf{s}_0 \rightarrow \mathbf{s}_1 \rightarrow \cdots \rightarrow \mathbf{s}_n)$, where $\mathbf{s}_n \in \mathcal{X}$ and each pair of consecutive states is related by an action, *i.e.*, $\forall i \, (\mathbf{s}_i, \mathbf{s}_{i+1}) \in \mathcal{A}$.

A *policy* on the MDP is a choice of distribution $P_F(\mathbf{s}'|\mathbf{s})$ for each $\mathbf{s} \in \mathcal{S} \setminus \mathcal{X}$ over the states $\mathbf{s}'$ reachable from $\mathbf{s}$ in a single action.[2] A policy induces a distribution over complete trajectories via

$$P_F(\mathbf{s}_0 \to \mathbf{s}_1 \to \cdots \to \mathbf{s}_n) = \prod_{i=0}^{n-1} P_F(\mathbf{s}_{i+1} \mid \mathbf{s}_i).$$

The marginal distribution over the final states of complete trajectories is denoted $P_F^\top$, a distribution on $\mathcal{X}$ that may in general be intractable to compute exactly, as $P_F^\top(\mathbf{x}) = \sum_{\tau \to \mathbf{x}} P_F(\tau)$, with the sum taken over all complete trajectories that end in $\mathbf{x}$.

A *reward function* is a mapping $\mathcal{X} \to \mathbb{R}_{>0}$, which is understood as an unnormalized probability mass on the set of terminal states (we will typically make the identification $R(\mathbf{x}) = \exp(-\mathcal{E}(\mathbf{x})/T)$, where $\mathcal{E} : \mathcal{X} \to \mathbb{R}$ is an energy function and $T > 0$ is a temperature parameter). The learning problem approximately solved by a GFlowNet is to fit a policy $P_F(\mathbf{s}'|\mathbf{s})$ such that the induced distribution $P_F^\top(\mathbf{x})$ is proportional to the reward function, *i.e.*,

$$P_F^\top(\mathbf{x}) \propto R(\mathbf{x}) = \exp(-\mathcal{E}(\mathbf{x})/T). \tag{1}$$

The policy $P_F(\mathbf{s}'|\mathbf{s})$ is parametrized as a neural network with parameters $\boldsymbol{\theta}$ taking $\mathbf{s}$ as input and producing the logits of transitioning to each possible subsequent states $\mathbf{s}'$. This problem is made difficult both by the intractability of computing $P_F^\top$ given $P_F$ and by the unknown normalization constant (partition function) on the right side of (1). Learning algorithms overcome these difficulties by introducing auxiliary objects into the optimization. Next, we review two relevant objectives.

**Detailed balance (DB)**   The DB objective (Bengio et al., 2023), requires learning two objects in addition to a parametric forward policy $P_F(\mathbf{s}'|\mathbf{s}; \boldsymbol{\theta})$ (we omit $\boldsymbol{\theta}$ when it causes no ambiguity):

- A *backward policy*, which is a distribution $P_B(\mathbf{s}|\mathbf{s}'; \boldsymbol{\theta})$ over the parents (predecessors) of any noninitial state in the MDP;
- A *state flow* function $F(\cdot; \boldsymbol{\theta}) : \mathcal{S} \to \mathbb{R}_{>0}$.

The detailed balance loss for a single transition $\mathbf{s} \to \mathbf{s}'$ is defined as

$$\ell_{\mathrm{DB}}(\mathbf{s}, \mathbf{s}'; \boldsymbol{\theta}) = \left( \log \frac{F(\mathbf{s}; \boldsymbol{\theta}) P_F(\mathbf{s}'|\mathbf{s}; \boldsymbol{\theta})}{F(\mathbf{s}'; \boldsymbol{\theta}) P_B(\mathbf{s}|\mathbf{s}'; \boldsymbol{\theta})} \right)^2. \tag{2}$$

The DB training theorem states that if $\ell_{\mathrm{DB}}(\mathbf{s}, \mathbf{s}'; \boldsymbol{\theta}) = 0$ for all transitions $\mathbf{s} \to \mathbf{s}'$, then the policy $P_F$ satisfies (1), *i.e.*, samples proportionally to the reward. The manner of selecting transitions $\mathbf{s} \to \mathbf{s}'$ on which to minimize (2) is discussed below.

The loss that performs best for the problems in this paper (see (6) in §3.3) is equivalent to DB with a particular parametrization of $\log F(s)$ that bootstraps learning by expressing the log-state flow as an additive correction to a *partially accumulated* negative energy.

**Trajectory balance (TB)**   The TB objective (Malkin et al., 2022) features a simpler parametrization: in addition to the action policy $P_F$, one learns a backward policy $P_B$ and only a single scalar $Z_{\boldsymbol{\theta}}$, an estimator of the partition function corresponding to the initial state flow $F(\mathbf{s}_0)$ in the DB parametrization. The TB loss for a complete trajectory $\tau = (\mathbf{s}_0 \to \mathbf{s}_1 \to \cdots \to \mathbf{s}_n = \mathbf{x})$ is

$$\ell_{\mathrm{TB}}(\tau; \boldsymbol{\theta}) = \left( \log \frac{Z_{\boldsymbol{\theta}} \prod_{i=0}^{n-1} P_F(\mathbf{s}_{i+1}|\mathbf{s}_i; \boldsymbol{\theta})}{R(\mathbf{x}) \prod_{i=0}^{n-1} P_B(\mathbf{s}_i|\mathbf{s}_{i+1}; \boldsymbol{\theta})} \right)^2. \tag{3}$$

The TB training theorem states that if $\ell_{\mathrm{TB}}(\tau; \boldsymbol{\theta}) = 0$ for all complete trajectories $\tau$, then the policy $P_F$ satisfies (1). Furthermore, $Z$ then equals the normalization constant of the reward, $\hat{Z} = \sum_{\mathbf{x} \in \mathcal{X}} R(\mathbf{x})$.

In practice, policies and flows are typically output in the log domain, *i.e.*, a neural network predicts logits of the distributions $P_F(\cdot|\mathbf{s})$, $P_B(\cdot|\mathbf{s}')$ and the log-flows $\log F(\mathbf{s})$ and $\log Z$.

---

[2]Note that one can unambiguously write the policy as a distribution over subsequent states and not over actions because the MDP is deterministic. Most GFlowNet work uses the equivalent language of directed acyclic graphs, see analysis in Pan et al. (2023b); we use the MDP formalism to be consistent with RL terminology and avoid clashes of notation and language with the graphs that are the inputs to CO problems.

**Training policy and exploration**   The DB and TB losses depend on individual transitions or trajectories, but leave open the question of how to choose the transitions or trajectories on which they are minimized. A common choice is to train in an on-policy manner, *i.e.*, rollout trajectories $\tau \sim P_F(\tau; \boldsymbol{\theta})$ and perform gradient descent steps on $\ell_{\text{TB}}(\tau; \boldsymbol{\theta})$ or on $\ell_{\text{DB}}(\mathbf{s}_i, \mathbf{s}_{i+1}; \boldsymbol{\theta})$ for transitions $\mathbf{s}_i \rightarrow \mathbf{s}_{i+1}$ in $\tau$. In this case, DB and TB have close connections to variational (ELBO maximization) objectives (Malkin et al., 2022).

However, an exploratory behaviour policy can also be used, for example, by sampling $\tau$ from a version of $P_F$ that is tempered or mixed with a uniform distribution (resembles $\epsilon$-greedy exploration in RL). Note that, unlike policy gradient methods in RL, GFlowNet objectives require no differentiation through the sampling procedure that yields $\tau$. The ability to stably learn from off-policy trajectories is a key advantage of GFlowNets over hierarchical variational models (Zimmermann et al., 2022; Malkin et al., 2023). See related study in Section C.

**Conditional GFlowNets**   The MDP and the reward function in a GFlowNet can depend on some conditioning information. For example, in the tasks we study, a GFlowNet policy sequentially constructs the solution to a CO problem on a graph $\mathbf{g}$, and the set of permitted actions depends on $\mathbf{g}$. The conditional GFlowNets we train achieve amortization by sharing the policy model between different $\mathbf{g}$, enabling generalization to $\mathbf{g}$ not seen in training.

## 2.2   Graph combinatorial optimization problems

We focus on the following four NP-hard CO problems on graphs: maximum independent set (MIS), maximum clique (MC), minimum dominating set (MDS), and maximum cut (MCut). A CO problem can be described with an undirected graph $\mathbf{g} = (V, E)$, where $V$ is the set of vertices and $E$ is the set of edges. Such problems typically require one to optimize over variables in a finite composite space to maximize or minimize some particular graph properties, as follows. Without loss of generality, we assume that all the graph weights equal one for simplicity, as our method can be easily extended to weighted graphs.

**Maximum independent set**   In graph theory, an independent set is a set of vertices $S$ in a given graph structure $\mathbf{g}$ where any pair of vertices $\{i, j\} \subseteq S$ are not neighbors *i.e.*, $\forall i, j \in S, (i, j) \notin E$. The MIS problem is to find such an independent set that has the largest possible size $|S|$.

**Maximum clique**   A clique is a subset of the vertices $S \subseteq V$ where all pairs of vertices are adjacent, *i.e.*, $\forall i, j \in S, (i, j) \in E$. The MC problem is to find a clique that has the largest size. We also remark that MC can be considered as a complementary problem of MIS, in the sense that the MC of any graph is actually the MIS of its complementary graph.

**Minimum dominating set**   A dominating set $S \subseteq V$ for a graph $\mathbf{g}$ is a subset of vertices such that, for any vertex in this graph, it is either in $S$, or it has a neighbor in $S$. The MDS problem is to find the smallest dominating set for a given graph structure.

**Maximum cut**   Given a subset of graph vertices $S \subseteq V$, the cut is defined as the number of edges between $S$ and $V \setminus S$. The MCut problem is to find a set of vertices $S$ that maximizes the cut.

## 3   Methodology

### 3.1   Optimization as probabilistic inference

A CO problem can be seen as a constrained energy minimization problem in a discrete composite space $\arg \min_{\mathbf{x} \in \mathcal{X}} \mathcal{E}(\mathbf{x})$, where $\mathcal{E}$ is the target energy function and $\mathcal{X}$ is the solution space or feasible set. This optimization problem can be considered roughly as sampling from an energy-based model $p_T^*(\mathbf{x}) \propto \exp\{-\mathcal{E}(\mathbf{x})/T\}$, where $T$ is the temperature parameter that controls the smoothness of the density landscape. Sampling from such highly structured space is non-trivial, therefore we propose to use GFlowNets to amortize this inference process. According to the GFlowNet theory, a perfectly trained GFlowNet with the reward function $R(\mathbf{x}) = \exp\{-\mathcal{E}(\mathbf{x})/T\}$ will be able to accurately sample from $p_T^*(\mathbf{x})$. In this way, GFlowNets trained with a reasonably small temperature $T$ can be used to search for solutions to given CO problems, as stated below.

**Proposition 1.** *Assume that the GFlowNet is perfectly trained for a given temperature $T$, i.e., the training loss over a policy with full support equals zero. Then, if $T \rightarrow \infty$, the distribution sampled*

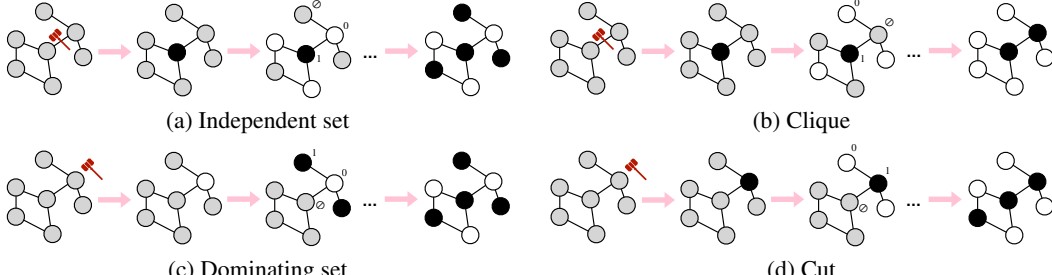

| (a) Independent set | (b) Clique |
| (c) Dominating set | (d) Cut |

Figure 2: Our proposed MDP designs for different CO tasks. All the tasks aim at a set of vertices as the problem solution, thus we use $0/1/\oslash$ to represent "not in the set" / "in the set" / "unspecified" for each vertex, which corresponds to white / black / gray in the figures. For each figure, the first arrow denotes conducting one action (marked with hammer), and the second arrow denotes the designed transition to guarantee any intermediate state represents a valid independent set / clique / dominating set / cut. The rightmost graph in each figure shows a feasible solution for the CO problem, where all vertices are specified (*i.e.*, belongs to the GFlowNet terminal state space $\mathcal{X}$).

*by GFlowNet will converge to a uniform distribution on $\mathcal{X}$; if $T \to 0$, the distribution sampled by GFlowNet will converge to the uniform distribution on optimal solutions to the optimization problem.*

Notice that each graph **g** corresponds to one unique sampling problem; thus, here we learn a graph conditional GFlowNet to amortize this condition distribution $p(\mathbf{x}|\mathbf{g})$, *i.e.*, every learnable GFlowNet component (like the forward / backward policy or the flow function) is conditioned on the graph **g**.

### 3.2 Designing Markov decision processes for GFlowNets

This section illustrates the design of appropriate Markov decision process (MDP) formulations for GFlowNet learning; see Figure 2 for a summary.

**State** For all the CO problems studied in this work, the solution **x** is a subset of vertices for a given graph, represented as a binary vector $\mathbf{x} = (\mathbf{x}^1, \ldots, \mathbf{x}^{|V|}) \in \mathcal{X} \subseteq \{0, 1\}^{|V|}$, where $\mathbf{x}^i = 1$ indicates that the $i$-th vertex belongs to the set and $\mathbf{x}^i = 0$ indicates that it does not. Note that the feasible solution space $\mathcal{X}$ is a subset of the full space $\{0, 1\}^{|V|}$, as some binary vectors encode a vertex set outside the feasible set (*e.g.*, a non-independent set in the MIS problem). The solution space is also the GFlowNet's terminal state space. The design of the GFlowNet state space is inspired by that in Zhang et al. (2022b). We begin by defining

$$\overline{\mathcal{S}} \triangleq \{(\mathbf{s}^1, \ldots, \mathbf{s}^{|V|}) : \mathbf{s}^d \in \{0, 1, \oslash\}, d = 1, \ldots, |V|\}, \tag{4}$$

where $\oslash$ represents an unspecified "void" or "yet unspecified" situation for a particular vertex. Notice that $\overline{\mathcal{S}}$ is a superset of the terminal state space $\mathcal{X}$. The initial state is the all-void vector $\mathbf{s}_0 = (\oslash, \oslash, \ldots, \oslash)$. In our MDP design, we only allow transforming vertex values from void ($\oslash$) to non-void (0 or 1) according to problem-specific rules. The trajectory terminates when all the entries are non-void, *i.e.*, every entry is either 0 or 1. Notice that we do not need to have an explicit stop action in the agent's action space.

Notice that not all vectors in $\overline{\mathcal{S}}$ that have no void entries lie in $\mathcal{X}$. Therefore, we must restrict $\mathcal{S}$ so that the set of terminal states is exactly the set of vectors encoding feasible solutions $\mathcal{X}$. To be precise, we define the state space $\mathcal{S}$ to be the set of states $\mathbf{s} \in \overline{\mathcal{S}}$ such that there exists at least one $\mathbf{x} \in \mathcal{X}$ that can be obtained from $\mathbf{s}$ by valid transitions.

Care must be taken to modify the set of permitted actions from each state accordingly, so that actions always produce states in $\mathcal{S}$. This turns out to be doable in the problems we study (as, for example, any subset of an independent set is independent). As an example, we next describe our action / transition / reward design for the MIS problem. Details for other tasks are deferred to the Appendix B.

**Action** The initial state is all-void, meaning that the partially constructed independent set is initialized with the empty set. The action of the MIS MDP is simply to choose one void vertex and add it to the current solution set by turning the entry value of the chosen vertex from $\oslash$ to 1.

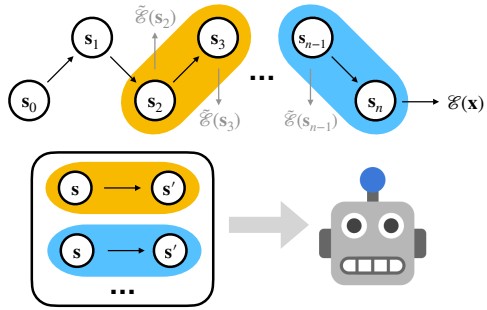

Figure 3: Illustration of transition-based GFlowNet training. We break complete trajectories into transitions, and then randomly choose some of the them to form a buffer to train the GFlowNet agent. We use $\mathbf{x}$ to denote the terminal state $\mathbf{s}_n$. $\tilde{\mathcal{E}}(\cdot)$ denotes designed intermediate learning signals.

**Transition** To ensure that the state can always be completed to an independent set, it is essential to carefully handle the actions taken. When a void vertex is chosen and its entry value is modified to 1, we also update the entry values of all its neighboring vertices to 0, which excludes the possibility of getting two adjacent vertices in the following steps. This proactive approach ensures that the independent set constraint is not violated in subsequent steps.

We remark that the feasible set $\mathcal{X}$ in this problem consists not of all independent sets, but of *order-maximal* independent sets, *i.e.*, those to which no vertex can be added while keeping them independent. Non-order-maximal independent sets cannot be constructed with such transitions.

**Reward** We set the log reward to be the resulting independent set size, *i.e.*, $\mathcal{E}(\mathbf{x}) = -|\mathbf{x}|_1$, where $|\mathbf{x}|_1$ denotes the $\ell_1$ norm, *i.e.*, the number of 1's in the binary vector $\mathbf{x}$.

### 3.3 Factors affecting training efficiency

**Transition-based GFlowNet training** Most successful GFlowNet implementations (*e.g.*, those in https://github.com/GFNOrg/) require a complete trajectory to compute the training loss and its gradient. Existing implementations with DB also specify the training loss at the trajectory level:

$$\mathcal{L}(\tau; \boldsymbol{\theta}) = \frac{1}{n}\sum_{t=0}^{n-1}\ell_{\text{DB}}(\mathbf{s}_t, \mathbf{s}_{t+1}; \boldsymbol{\theta}), \quad \tau = (\mathbf{s}_0, \mathbf{s}_1, \ldots, \mathbf{s}_n), \tag{5}$$

and calculate the parameter gradient update with $\mathbb{E}_{\tau\sim\pi(\tau)}\left[\nabla_{\boldsymbol{\theta}}\mathcal{L}(\tau; \boldsymbol{\theta})\right]$ with some potentially off-policy distribution $\pi(\tau)$. This is also the case for other GFlowNet algorithms, such as flow matching (Bengio et al., 2021) and subtrajectory balance (Madan et al., 2023).

These implementations work well for moderate-scale trajectories, as previous GFlowNet works have shown (Malkin et al., 2022; Madan et al., 2023). However, for very long trajectories, such a design hinders efficient training: for a single complete trajectory which contains $n$ transitions, one needs $n$ calls of the neural network forward passes and storing all the intermediate feature maps and parameters in the GPU memory. Each forward pass contains multiple message passing operations on given graph structures, which is computationally expensive (linearly increasing with $n$) in terms of speed and memory storage. In our experiments, we encounter large-scale problems where the trajectory is as long as $\sim 400$ in length; nonetheless, with such graphs, our adopted graph neural networks can only support

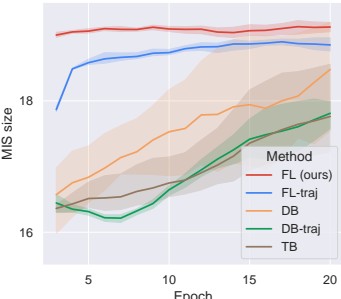

Figure 4: Comparison between different GFlowNet variants.

forward and backward passes with batch size approximately 250 (which is much smaller than 400) on a 40 GB GPU memory device. This prohibits the usage of these trajectory-based GFlowNet training objectives on large-scale graph applications. In addition, the correlation between consecutive samples may incur stability issues (Mnih et al., 2015).

To this end, we turn to use a transition-based GFlowNet training approach wihtout the knowledge of complete trajectories, which is first proposed in Deleu et al. (2022) and has shown effectiveness in Nishikawa-Toomey et al. (2022); Deleu et al. (2023). We randomly sample $B$ transitions $\mathcal{B} = \{(\mathbf{s}^b, \mathbf{s'}^b)\}_{b=1}^{B}$ from a complete trajectory and construct detailed balance-based loss: $\mathcal{L}(\boldsymbol{\theta}) = \frac{1}{B}\sum_{b=1}^{B} \ell(\mathbf{s}^b, \mathbf{s'}^b; \boldsymbol{\theta})$. This enables GFlowNets to be efficiently trained with long trajectories and limited GPU memory, which also converges faster. A corresponding schematic illustration of the algorithm can be found in Figure 3. In Figure 4, we show a comparison of learning efficiency between transition-based and trajectory-based GFlowNet implementations, where "-traj" denotes the latter variant. Different methods here share similar speed for one epoch training, thus from the figure we can see that transition-based approaches learn more efficiently. Besides the improved computation efficiency, our transition-based approach is also more memory efficient – as the memory occupation is proportional to the batch size instead of trajectory lengths (as for trajectory-based implementation), demonstrating its applicability to long-horizon problems.

**Improving credit assignment with intermediate learning signals**    For normal GFlowNet training methods, the only learning signal in a trajectory comes from the terminal states and their associated reward values. This results in a relatively slow credit assignment process, and is inefficient to propagate information from near-terminal states to the early states due to the difficulty of attributing credits of each action in a long trajectory. The ability to learn from incomplete trajectories is especially important in our proposed transition-based training, since we are not using all the transitions from the complete trajectories. Therefore, we incorporate intermediate learning signals via the forward-looking (Pan et al., 2023a, FL) technique:

$$\ell_{\mathrm{FL}}(\mathbf{s}, \mathbf{s}'; \boldsymbol{\theta}) = \left(-\tilde{\mathcal{E}}(\mathbf{s}) + \log \tilde{F}(\mathbf{s}; \boldsymbol{\theta}) + \log P_F(\mathbf{s}'|\mathbf{s}; \boldsymbol{\theta}) + \tilde{\mathcal{E}}(\mathbf{s}') - \log \tilde{F}(\mathbf{s}'; \boldsymbol{\theta}) - \log P_B(\mathbf{s}|\mathbf{s}'; \boldsymbol{\theta})\right)^2, \quad (6)$$

where $\tilde{\mathcal{E}}(\cdot) : \mathcal{S} \to \mathbb{R}$ is a continuation of the reward energy $\mathcal{E}(\cdot) : \mathcal{X} \to \mathbb{R}$ which is only defined in the terminal state space $\mathcal{X}$. For simplicity, here we ignore the conditioning on graph structure $\mathbf{g}$. This FL method enables dense supervision signals to GFlowNet training, resulting in faster credit assignment as can be seen in Figure 3. Notice here that we need to design a handcrafted reward $\tilde{\mathcal{E}}(\mathbf{s})$ for all possible latent states to reflect our estimation on intermediate signals. For MIS problems, we naturally define $-\tilde{\mathcal{E}}(\mathbf{s})$ to be the number of vertices in the current set, *i.e.*, the number of 1 entries in state $\mathbf{s}$. This semantically coincides with the definition of an MIS terminal reward. We defer the intermediate reward design for other tasks to the Appendix. The effectiveness of FL against the DB or TB algorithms can be seen in Figure 4. We summarize the resulting algorithm in Algorithm 1.

## 4    Related work

**GFlowNets**    GFlowNets were intended as diversity-seeking samplers for biological sequence and molecule design, an application area that continues to motivate research (Bengio et al., 2021; Jain et al., 2022, 2023b; Shen et al., 2023; Jain et al., 2023a). However, much recent work has used GFlowNets as samplers for Bayesian posterior distributions, *e.g.*, causal discovery (Deleu et al., 2022; Nishikawa-Toomey et al., 2022; Atanackovic et al., 2023), amortized variational EM with discrete latents (Hu et al., 2023), neurosymbolic inference (van Krieken et al., 2022), and feature attribution in classifiers (Li et al., 2023). The theory and optimization techniques for GFlowNets have also evolved, with improved training objectives and exploration techniques (Malkin et al., 2022; Madan et al., 2023; Pan et al., 2022, 2023a; Shen et al., 2023), better understanding of their connections to variational methods (Zhang et al., 2022a; Malkin et al., 2023; Zimmermann et al., 2022), and extensions to stochastic (Zhang et al., 2023c; Pan et al., 2023b) and continuous (Lahlou et al., 2023) MDPs.

**ML for combinatorial optimization**    The surge of machine learning for CO problems alleviates the reliance on hand-crafted heuristics while enabling generalization to new instances (Bengio et al., 2018; Cappart et al., 2021). Some methods (Li et al., 2018; Gasse et al., 2019; Gupta et al., 2020; Sun & Yang, 2023) rely on supervised information from expert solvers, which can be hard to obtain. Alternative approaches that leverage reinforcement learning (Dai et al., 2017; Kool et al., 2019; Chen & Tian, 2019; Yolcu & Póczos, 2019; Ahn et al., 2020; Delarue et al., 2020; Drori et al., 2020) or other unsupervised learning objectives (Karalias & Loukas, 2020; Sun et al., 2022; Wang et al., 2022) broaden the applicability of learning for CO problems. However, the mode-collapse issue might hinder the diversity and thus the solution coverage. In this regard, GFlowNets are easy to train while also designed for discovering multiple modes. As the first example to show the advantage of

Table 1: Max independent set experimental results. We report the absolute performance, approximation ratio (relative to KAMIS), and inference time. All algorithms fall into three categories: OR (operations research), SL (supervised learning), and UL (unsupervised learning). "—" denotes no reasonable result is achieved by the corresponding algorithm in 10 hours. Time shown as H:M:S.

| METHOD | TYPE | SMALL | | | LARGE | | | SATLIB | | |
|--------|------|-------|-------|-------|-------|-------|-------|--------|-------|-------|
| | | SIZE ↑ | DROP ↓ | TIME ↓ | SIZE ↑ | DROP ↓ | TIME ↓ | SIZE ↑ | DROP ↓ | TIME ↓ |
| GUROBI | OR | 19.98 | 0.01% | 47:34 | 40.90 | 5.21% | 2:10:26 | 425.95 | 0.00% | 3:43:19 |
| KAMIS | OR | 20.10 | 0.00% | 1:24:12 | 43.15 | 0.00% | 2:03:36 | 425.96 | 0.00% | 4:15:41 |
| PPO | UL | 19.01 | 5.42% | 1:17 | 32.32 | 25.10% | 7:33 | 421.49 | 1.05% | 13:12 |
| INTEL | SL | 18.47 | 8.11% | 13:04 | 34.47 | 20.12% | 20:17 | — | — | — |
| DGL | SL | 17.36 | 13.61% | 12:47 | 34.50 | 20.05% | 23:54 | — | — | — |
| OURS | UL | **19.18** | 4.57% | 0:32 | **37.48** | 13.14% | 4:22 | **423.54** | 0.57% | 23:13 |

GFlowNets in CO problems, Zhang et al. (2023a) tackles the robust job scheduling problem that is central to compiler optimization. In our paper, we formalize CO under the GFlowNet framework with principled MDP design and demonstrate its effectiveness in a wide range of graph CO tasks.

## 5 Experiments

We conduct extensive experiments on various graph CO tasks to demonstrate the effectiveness of the proposed GFlowNet approach. For MIS problems, we follow the setup in the MIS benchmark from Böther et al. (2022), while for other tasks, we follow the experimental setup in Sun et al. (2022).

**Datasets** As Dai et al. (2021) have pointed out that problems in existing synthetic graph data are relatively easy for MIS and MC, we take the more complicated RB graphs (Xu & Li, 2000) following Karalias & Loukas (2020). For realistic data, we take the SATLIB dataset (Hoos et al., 2000), which is reduced from SAT instances in conjunctive normal form. For the other two tasks, namely MDS and MCut, we adopt BA graphs (Barabási & Albert, 1999) following Sun et al. (2022). For all types of synthetic graph data, we generate two scales of datasets (denoted SMALL and LARGE), which respectively contain around 200 to 300 vertices and 800 to 1200 vertices.

**Baselines** For MIS problems, we compare with the baselines in the MIS benchmark. For classical operation research (OR) methods, we include a general-purpose mixed-integer program solver (GUROBI) and a MIS-specific solver (Lamm et al., 2017, KAMIS). For learning-based methods, we compare with a reinforcement learning-based PPO method (Ahn et al., 2020), and supervised learning with tree search refinement, in two different implementations (Li et al. (2018, INTEL) and Böther et al. (2022, DGL)). For the non-MIS tasks, we compare with the GUROBI solver, two heuristic methods which are greedy and mean-field annealing (Bilbro et al., 1988, MFA), and two state-of-the-art probabilistic methods (Karalias & Loukas, 2020) and the annealed version (Sun et al., 2022). We use ERDOS and ANNEAL to denote these two learning-based methods. For max-cut problems, we also adopt a semi-definite programming baseline (coined SDP in the result tables) which aims at a relaxation of the MCut task. We set its maximal running time to 10 hours. For the methods already contained in the MIS benchmark, we train them from scratch and report performance with the same protocol; for other algorithms, we write our own implementations and test them in the same way.

**Results & analysis** We evaluate both the performance and the inference time and report the mean value of the objective (*e.g.*, set size in MIS) and the approximation ratio relative to the best-performing non-ML solver, treated as an oracle.[3] The time denotes the total latency of evaluating on the test set. The best results among non-OR methods are marked in bold. MIS results are demonstrated in Table 1, where the problem-specific solver KAMIS is served as the gold standard for calculating the drop. For MIS methods, the larger the size of the independent set found, the better the algorithm. The "UL" (unsupervised learning) refers to the algorithms that do not need labels (*i.e.*, solutions found by solvers) in the training set. For SATLIB, none of the supervised learning baselines can

---

[3]The approximation ratio is computed on the total cost aggregated over the test set and is defined as the DROP $(1 - \frac{\text{algorithm}}{\text{oracle}})$ for maximization problems and the GAP $(1 - \frac{\text{oracle}}{\text{algorithm}})$ for minimization problems.

Table 2: Max clique, min dominating set, and max cut results on small graphs ($|V|$ between 200 and 300). We report absolute performance, approximation ratio, and inference time. All algorithms fall into three categories: OR (operations research), H (heuristic), and UL (unsupervised learning). The time latency is shown in the form of hour:minute:second.

| METHOD | TYPE | MC | | | MDS | | | MCUT | | |
|---|---|---|---|---|---|---|---|---|---|---|
| | | SIZE ↑ | DROP ↓ | TIME ↓ | SIZE ↓ | GAP ↓ | TIME ↓ | SIZE ↑ | DROP ↓ | TIME ↓ |
| GUROBI | OR | 19.05 | 0.00% | 1:55 | 27.89 | 0.00% | 1:47 | 732.47 | 0.00% | 13:04 |
| SDP | OR | — | — | — | — | — | — | 700.36 | 4.38% | 35:47 |
| GREEDY | H | 13.53 | 28.98% | 0:25 | 37.39 | 25.41% | 2:13 | 688.31 | 6.03% | 0:13 |
| MFA | H | 14.82 | 22.15% | 0:27 | 36.36 | 23.29% | 2:56 | **704.03** | 3.88% | 1:36 |
| ERDOS | UL | 12.02 | 36.90% | 0:41 | 30.68 | 9.09% | 1:00 | 693.45 | 5.33% | 0:46 |
| ANNEAL | UL | 14.10 | 25.98% | 0:41 | 29.24 | 4.62% | 1:01 | 696.73 | 4.88% | 0:45 |
| OURS | UL | **16.24** | 14.75% | 0:42 | **28.61** | 2.52% | 2:20 | **704.30** | 3.85% | 2:57 |

Table 3: Max clique, min dominating set, and max cut results on large graphs (whose $|V|$ is between 800 and 1200). We use the same format as Table 2.

| METHOD | TYPE | MC | | | MDS | | | MCUT | | |
|---|---|---|---|---|---|---|---|---|---|---|
| | | SIZE ↑ | DROP ↓ | TIME ↓ | SIZE ↓ | GAP ↓ | TIME ↓ | SIZE ↑ | DROP ↓ | TIME ↓ |
| GUROBI | OR | 33.89 | 0.00% | 16:40 | 103.80 | 0.00% | 13:48 | 2915.29 | 0.00% | 1:05:29 |
| SDP | OR | — | — | — | — | — | — | 2786.00 | 4.43% | 10:00:00 |
| GREEDY | H | 26.71 | 21.17% | 0:25 | 140.52 | 26.13% | 35:01 | 2761.06 | 5.29% | 3:07 |
| MFA | H | 27.94 | 17.56% | 2:19 | 126.56 | 17.98% | 36:31 | 2833.86 | 2.79% | 7:16 |
| ERDOS | UL | 25.43 | 24.96% | 2:16 | 116.76 | 11.10% | 3:56 | **2870.34** | 1.54% | 2:49 |
| ANNEAL | UL | 27.46 | 18.97% | 2:16 | 111.50 | 6.91% | 3:55 | 2863.23 | 1.79% | 2:48 |
| OURS | UL | **31.42** | 7.29% | 4:50 | **110.28** | 5.88% | 32:12 | 2864.61 | 1.74% | 21:20 |

achieve meaniningful results (average MIS size $< 400$) within 10 hours, thus we use "—" to mark performance. One can see that GFlowNets surpass all the learning-based baselines in the sense of achieving the largest independent set solution.

For the MC & MDS & MCut problems, we exhibit the experimental results on small graphs in Table 2 and results on large graphs in Table 3, where GUROBI serves as the gold standard to calculate the performance drop ratio. Notice that for MDS problems, the smaller the vertex size result, the better the performance, which is opposite to other tasks. We observe that GFlowNet outperforms other baselines across different tasks and problem scales, with the only exception being the large-scale max-cut problem. This reflects the fairly universal effectiveness of the proposed method. We also notice that GFlowNets require a longer inference time compared to other methods. This is because the GFlowNet only adds one vertex into the set at each step, thus it needs multiple steps to output a vertex set solution. In contrast, probabilistic methods such as "Erdős goes neural" (Karalias & Loukas, 2020) only require one neural network forward pass during inference time to give a coarse estimate of the solution. We can see how this compares with a sequential generation approach as in GFlowNets, where each decision is taken in the context of the previously already taken decisions, ensuring a better coordinated set of decisions.

**Ablation study** We now conduct in-depth ablation studies to evaluate the key design choices in our method. We first study the difference between a series of GFlowNet variants in Figure 4, based on the MIS task with small scale graph data. We compare transition-based FL, trajectory-based FL, transition-based DB, trajectory-based DB, and TB (which only has trajectory-based implementation). Our result indicates that GFlowNet's transition-based FL implementation learns the fastest for CO tasks, which supports our modeling choice in Section 3.3. Figure 5 in Section C summarizes additional ablation studies including the temperature annealing, off-policy exploration strategy, and network architectures. We vary the GFlowNet's temperature coefficient which scales the temperature hyperparameter; we ablate the off-policy exploration during the rollout stage of GFlowNet training; we also ablate the GFlowNet architecture. Our results indicate that the GFlowNet is robust to a

wide range of hyperparameters, making it appealing to be applied to different CO problems, which validates the effectiveness of our proposed methodology from another perspective.

## 6 Conclusion

Our work focuses on the challenges of solving CO problems using unsupervised learning approaches. This work contributes to this growing field by proposing to apply the principled GFlowNet decision-making framework to CO tasks. By combining the power of probabilistic inference and sequential decision-making, GFlowNets offer a promising direction for finding a diverse set of high-quality candidate solutions in CO problems. Technically, we have developed problem-specific MDPs and efficient learning algorithms to address the challenges associated with learning from long trajectories, making our approach practical and scalable in large-scale setups. Our extensive numerical results showcase the effectiveness and efficiency of GFlowNets in solving NP-hard level CO problems, highlighting their ability to generate a set of diverse high-quality solutions. We believe that our work opens up new possibilities for addressing the limitations of existing approaches and paves the way for future research at the intersection of machine learning and combinatorial optimization.

## Acknowledgement

The authors are thankful to Yuandong Tian, David Wei Zhang, Haoran Sun, Zhiqing Sun, Taoan Huang, Sophie Xhonneux, and Zhaoyu Li. Yoshua acknowledges funding from CIFAR, NSERC, Intel and Samsung. Aaron is supported by Hitachi, Samsung, Canadian Research Chair and CIFAR. Dinghuai acknowledges Daozhen Lin for being his tour guide during their visit to beautiful Italy.

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

# A Notations

| Symbol | Description |
| --- | --- |
| $\mathcal{S}$ | state space |
| $\mathcal{X}$ | object (terminal state) space, subset of $\mathcal{S}$ |
| $\mathcal{A}$ | action / transition space (edges $\mathbf{s} \to \mathbf{s}'$) |
| $\mathbf{s}$ | state in $\mathcal{S}$ |
| $\mathbf{s}_0$ | initial state, element of $\mathcal{S}$ |
| $\mathbf{x}$ | terminal state in $\mathcal{X}$ |
| $\tau$ | complete trajectory |
| $\boldsymbol{\theta}$ | learnable parameter of GFlowNets |
| $F : \mathcal{S} \to \mathbb{R}$ | state flow |
| $P_F$ | forward policy (distribution over children) |
| $P_B$ | backward policy (distribution over parents) |
| $R : \mathcal{X} \to \mathbb{R}_{>0}$ | reward function (unnormalized target density) |
| $Z$ | scalar, equal to $\sum_{\mathbf{x} \in \mathcal{X}} R(\mathbf{x})$ |
| $\mathcal{E} : \mathcal{X} \to \mathbb{R}$ | energy function, equal to $-T * \log R(\mathbf{x})$ |
| $\tilde{\mathcal{E}} : \mathcal{S} \to \mathbb{R}$ | energy in state space, to provide intermediate learning signals |
| $T$ | scalar, temperature |
| $\mathbf{g}$ | graph configuration |
| $V$ | vertex set of graph $\mathbf{g}$ |
| $S$ | subset of $V$, solution to some CO problem |
| $E$ | edge set of graph $\mathbf{g}$ |

# B MDP designs

In all MDPs, the agent maintain a set of vertices as the state. This is achieved by assign a value among $\{0, 1, \oslash\}$ for each of the vertex, which corresponds to "not in the set" / "in the set" / "unspecified". At each step, the agent performs an action, which is essentially changing the entry value of one vertex from $\oslash$ to a specified binary value. After that, the designed transition may modify the value of other unspecified vertices to ensure the state will represent a feasible solution. The agent will receive a terminal reward when no action can be taken (*i.e.*, termination). An illustration of the following descriptions can be found in Figure 2.

## B.1 Maximum clique MDP

Here we think of the all-void initial state means that we have an empty set of vertex at the beginning.

**Action** The action is to choose an unspecified vertex and include it in the current set. This is changing the entry value of that vertex in the state vector representation from $\oslash$ to $1$.

**Transition** We need to ensure that the current set always represents a clique, thus it is crucial to enforce the connectivity between vertices in the set. If a vertex could potentially join the clique in the future, then it must connect to all the vertices in the current set. Hence, after performing an action, we identify the vertices that fail to meet this requirement and mark them as $0$. This ensures that the clique constraint is preserved throughout the generation process.

**Reward** We set the log reward to be the resulting clique size, *i.e.*, $\mathcal{E}(\mathbf{x}) = -|\mathbf{x}|_1$. For intermediate state $\mathbf{s}$, we define its intermediate signal by setting $-\tilde{\mathcal{E}}(\mathbf{s})$ to be the number of $1$ in $\mathbf{s}$.

## B.2 Minimum dominating set MDP

In MDS, we start with an all-void initial state, where all vertices (which are in unspecified state) are considered to *in* the target dominating set. This is different from other MDP designs, where we think of unspecified void state as *not in* the vertex set. In this representation, the entire vertex set $V$ forms a trivial dominating set. Starting from this, we perform actions to remove vertices from the current set but keep it to be a dominating set all the time.

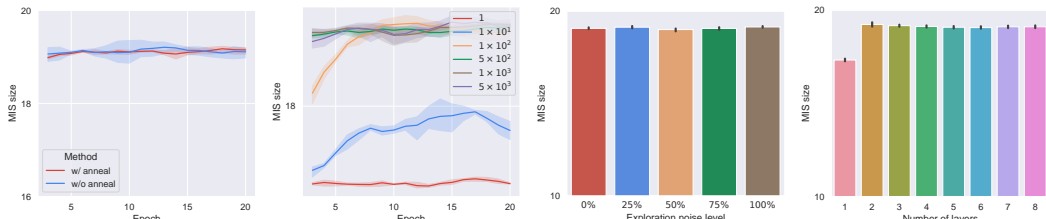

Figure 5: Ablation study on GFlowNet training. These experiments are conduct on the small scale MIS task. From left to right: ablation on whether to use temperature annealing, inverse temperature, how much percentage of uniform noise used in off-policy exploration, and number of message passing layers in the graph neural network. These results demonstrate the robustness of GFlowNets against different hyperparameter setups.

**Action**   Each action corresponds to selecting a vertex and *removing* it from the current dominating set. This is done by changing the entry value of the chosen vertex from $\oslash$ to 0.

**Transition**   For a vertex in the graph, if it is adjacent to the current set and all of its neighbors are also connected to the set, removing this vertex from the set will not violate the dominating set requirement. For other isolated vertices, we change their values to 1 to ensure they stay in the set, thus the state $\mathbf{s}$ consistently representing a dominating set. By making these adjustments, we maintain the integrity of the dominating set representation throughout the process.

**Reward**   The log reward is set to be the *negative* dominating set size, *i.e.*, $\mathcal{E}(\mathbf{x}) = |\mathbf{x}|_1$. For intermediate state $\mathbf{s}$, we define its intermediate signal by setting $\tilde{\mathcal{E}}(\mathbf{s})$ to be the number of 1 in $\mathbf{s}$.

### B.3   Maximum cut MDP

We start with an empty solution vertex set, *i.e.*, all the vertices are in $\oslash$ value.

**Action**   Same as MIS and MC, the action is to add one vertex to the current solution set of vertices, by changing its entry value from $\oslash$ to 1.

**Transition**   After each action is carried out, we perform the following check: if involving one void vertex to the solution vertex set would decrease the cut value, then we exclude this vertex by labeling it to 0 from void $\oslash$. This guarantees that when the trajectory terminates (*i.e.*, no void unspecified vertex), the cut is locally maximal.

**Reward**   We set the negative energy to be the size of the current cut set, *i.e.*, the number of edges between all 1 vertices and all 0 vertices. For intermediate state $\mathbf{s}$, we define its intermediate signal by setting $-\tilde{\mathcal{E}}(\mathbf{s})$ to be the number of edges between all 1 vertices and other vertices (note that we think of void vertices as not in the solution set).

## C   GFlowNet details

We start with the proof of Proposition 1 and then describe other algorithmic details.

*Proof.* According to the theory of GFlowNets (Bengio et al., 2023), if GFlowNet's training loss is zero on full support, then the GFlowNet is able to sample correctly from the target distribution $p_T^*(\mathbf{x})$. When we gradually move temperature $T$ from finite to $\infty$, the distribution $p_T^*(\mathbf{x})$ will gradually become the uniform distribution over $\mathcal{X}$, thus the distribution sampled by GFlowNet will converge to this uniform distribution; when the temperature gradually approaching zero, the distribution $p_T^*(\mathbf{x})$ will gradually become the uniform distribution over all argmax solution of $\arg\max_{\mathbf{x}} \mathcal{E}(\mathbf{x})$, thus the distribution sampled by GFlowNet will converge to this uniform distribution. $\square$

For the GFlowNet architecture, we use graph isomorphism network (Xu et al., 2019, GIN) with 5 hidden layers and 256 dimensional hidden size, for both the forward policy and the state flow function. We double the number of hidden layers for SATLIB experiment. The input of the GIN is a integer

Table 4: Max independent set experimental results on ER graph data in the same form as Table 1.

| METHOD | TYPE | ER-700-800 | | | ER-9000-11000 | | |
|--------|------|--------|--------|---------|--------|--------|---------|
| | | SIZE ↑ | DROP ↓ | TIME ↓ | SIZE ↑ | DROP ↓ | TIME ↓ |
| GUROBI | OR | 43.47 | 3.33% | 2:10:05 | — | — | — |
| KAMIS | OR | 44.98 | 0.00% | 2:11:21 | 374.57 | 0.00% | 7:37:21 |
| PPO | UL | 41.11 | 8.60% | 4:30 | 344.67 | 7.98% | 1:02:29 |
| INTEL | SL | 37.34 | 16.99% | 28:32 | — | — | — |
| DGL | SL | 38.71 | 13.94% | 29:36 | — | — | — |
| OURS | UL | 41.14 | 8.53% | 2:55 | 349.42 | 6.98% | 1:49:43 |

vector represented state in the form of $\{0, 1, 2\}^{|V|}$ followed by an embedding layer, where we use 2 to denote the unspecified void $\oslash$. For the forward policy GIN, we set a unidimensional output node feature followed by a softmax operation over all vertices, which gives a categorical distribution on all the vertices. For the state flow GIN, we use a graph pooling layer to extract a unidimensional graph-level output to serve as the flow value. We fix the backward policy to be a uniform distribution over all chosen vertices for simplicity. Theoretically, we can only use one GIN for both the flow function and the forward policy, but we find it is much less stable. We hypothesize that *the flow function training stability is very important in DB-based GFlowNet methods*, and that *most previous GFlowNet works underfit the flow function and underestimate the performance of DB.*

We use the Adam optimizer with default $1 \times 10^{-3}$ learning rate without hyperparameter tuning. After breaking the complete trajectories into transition pieces, we combine them to many buffer with batchsize equals 64 to calculate forward-looking based detailed balance loss. The training keeps 20 epochs, although most experiments actually converge in the first 5 epochs. For the SATLIB experiment, our proposed method converges within the first epoch. During the inference phase, we simply the same evaluation protocol in Ahn et al. (2020). To be concrete, we sample 20 solutions for each graph configuration and take the best result.

We conduct a series of ablation study with small graph MIS experiments in Figure 5. Each algorithm is repeated with five different random seeds. For the temperature $T$ used in GFlowNet's reward shaping, we try values of the inverse temperature from 1 to 5000 and find values higher or equal than 100 will obtain similar results. We thus simply use an inverse temperature of 500 for all the tasks. We also use temperature annealing, to start with temperature of 1 and terminate with the target temperature level (although the ablation study shows that GFlowNet could reach same performance without the annealing trick). During the training, GFlowNet will need to rollout (*i.e.*, interact with the environment to collect complete trajectories) to collect training data. In this work, we do on-policy exploration, which means the policy used in the rollout stage is the current forward policy. Nonetheless, GFlowNet has the ability of doing off-policy exploration. In the third ablation study, we mix the forward policy with different levels of uniform noise distribution, and find that GFlowNet is very robust under this off-policy setups. In this fourth ablation study, we change the number of hidden layer in the GIN architecture and find GFlowNet is also robust to this hyperparameter.

## D    More about experiments

For BA graphs, we use the variants with 4 attaching edges. For small scale datasets, we first sample the number of vertices uniformly from 200 to 300 and generate the graph instances. For large-scale problems, it is uniformly sampled from 800 to 1200. Both the small and large synthetic graph trainsets contain 4000 data points for either RB graph or BA graph. For the SATLIB dataset, there are 40000 data points in total, where the largest graph has 1347 vertices. For all datasets, we use a validation set and a test set, both with 500 data points. We choose the hyperparameters of each algorithm based on the validation performance. Regarding hardware, we use NVIDIA Tesla V100 Volta GPUs. We repeat the experiments for five different random seeds.

We also conduct MIS experiments on Erdos-Renyi (ER) graph data shown in Table 4. We use ER data as large as $700 \sim 800$ vertices and $9000 \sim 11000$ vertices for a comparison with Qiu et al. (2022, DIMES) and Sun & Yang (2023, DIFUSCO). The training and evaluation protocols are not exactly the same, but we still present results from these baselines for completeness. DIMES achieves 38.24/42.06 and 320.50/332.80 ("/" means two variants of their algorithm) respectively

for these two tasks; DIFUSCO achieves 38.83/41.12 ("/" means two variants of their algorithm) for the ER-700-800 problem, and is not scalable for extremely large problems like ER-9000-11000. The difference in protocol includes: GFlowNet uses less training data samples; DIFUSCO is a supervised learning algorithm (requires a solver to provides solution), while DIMES and GFlowNet are unsupervised methods; the test data may not be the same (they generate their own fold). For GFlowNet in the largest task, we use a temperature of 1000 and a batch size of 32. In summary, we can see that GFlowNet also achieves state-of-the-art performance in this setup, especially in large problems.

In MIS benchmark, for the MIS supervised learning baselines (INTEL and DGL), we disable the usage of the graph reduction and 2-opt local search tricks, but only adopt the queue pruning and weighted queue pop technique. This is because even random initialized neural network can be used to find near optimal solutions with graph reduction and local search as pointed out by Böther et al. (2022). Furthermore, these forbidden techniques are very time-consuming, making the inference time exceeding our time limit constraint. For KAMIS, we use the redumis algorithm. For GUROBI, we formulate MIS as quadratic programming problem and then use the pygurobi python package to solve it. We also run Erdos-based probabilistic methods (ERDOS and ANNEAL) for MIS, but their results are worse than any baselines methods in the MIS benchmark, thus we do not include them in Table 1. In the max cut task, we use the cvxpy python package to implement the SDP baseline.

