# OpenReview forum: "Let the Flows Tell:  Solving Graph Combinatorial Problems with GFlowNets"
_NeurIPS.cc/2023/Conference — NeurIPS 2023 spotlight_

### Official Review · Reviewer_PEdA · 2023-07-04

**Soundness:** 4 excellent
**Presentation:** 3 good
**Contribution:** 4 excellent
**Rating:** 8
**Confidence:** 4

**Summary:**

This paper suggests a new graph combinatorial optimization method by leveraging a generative flow network (GFlowNet) and other novel technics to improve it. Maximum independent sets (MIS) and their variants are very important problems as they can be applied to several high-impact tasks such as network communication. As these problems are, NP-hard exact method cannot solve them within a reasonable time, so that approximated solver such as a deep learning policy can support the production of near-optimal solutions quickly. This work follows this research trends and makes state-of-the-art over the deep learning baselines.

A major contribution is the usage of GFlowNet rather than PPO because exploration of GFlowNet can cover the symmetric nature lies in combinatorial solution space using DAG-based MDP construction. The novelty of this work is the improvement of GFlowNet for long trajectory generation of combinatorial optimization, so this work proposed (1) transition-based training and (2) intermediate learning signal training, which is a simple yet intuitive idea.

While this work is concrete and interesting, I have several questions and concerns; please address these concerns in the rebuttal discussion phase.

I just wrote the questions below.

**Strengths:**

This work is novel, and its performance seems to be promising. First, they cleverly leverage existing Markov decision process formulation of learning what to defer (LwD), as the state is described as $s \in \{0,1,2\}$ where 0 is included, one is excluded, and two is deferred; please make clear spotlight that this MDP idea is from prior work of LwD [1].

Second, they leveraged GFlowNet rather than PPO so that it could consider the symmetric nature of combinatorial solution space.

Third, their additional techniques are widely studied things in GFlowNet research (e.g., the intermediate reward is similar to Generative Augmented Flow Network [2], and the transition-based method is identical to sub-trajectory balance [3]) but made intuitive variations for combinatorial optimization.

[1] Ahn, Sungsoo, Younggyo Seo, and Jinwoo Shin. "Learning what to defer for maximum independent sets." International Conference on Machine Learning. PMLR, 2020.

[2] Pan, Ling, et al. "Generative augmented flow networks." arXiv preprint arXiv:2210.03308 (2022).

[3] Madan, Kanika, et al. "Learning GFlowNets from partial episodes for improved convergence and stability." arXiv preprint arXiv:2209.12782 (2022).

**Weaknesses:**

One can be said that technical novelty is limited as every technique are already actively explored; I think making the novel combination of existing technics is also significant.

I think the description of many ideas follows existing trends of GFlowNet; I enjoyed reading this paper, but maybe some who don't know about GFlowNet work can miss many detailed parts of this technique. Please make an explicit description of each technique and give a reference idea of where you are inspired by and what is the major difference, e.g., sub-trajectory balance vs. transition-based learning (assume that reader also not familiar with the sub-trajectory balance).

Finally, there are several neural combinatorial optimizations works [1,2] that study the symmetric nature of combinatorial optimization. I humbly suggest to include as a reference paper.

[1] Kim, Minsu, Junyoung Park, and Jinkyoo Park. "Sym-nco: Leveraging symmetricity for neural combinatorial optimization." Advances in Neural Information Processing Systems 35 (2022): 1936-1949.

[2] Kim, Hyeonah, et al. "Symmetric Exploration in Combinatorial Optimization is Free!." arXiv preprint arXiv:2306.01276 (2023).

**Questions:**

1. How does diversity (e.g., the edit distance between sampled solution) changes when your new techniques are applied? (to compare with another method, including the trajectory balance method)?

2. What kinds of post-processing methods are used for solution generation? Did you samples multiple samples from the learned sampler? Then what is the actual sampling width? How about applying local search just as learning what to defer (LwD) [1] ?

3. To compare with LwD, did you put 2opt for LwD following their original paper?

4. Can this method extend to other combinatorial optimization, such as the traveling salesman problem (TSP)?

[1] Ahn, Sungsoo, Younggyo Seo, and Jinwoo Shin. "Learning what to defer for maximum independent sets." International Conference on Machine Learning. PMLR, 2020.


**Limitations:**

GFlowNet is an on-going framework having a lot of limitations. Please make explicit limitations on the main paper for future researchers.

---

> ### Author Rebuttal · Authors · 2023-08-09
>
> > Please make an explicit description of each technique and give a reference idea of where you are inspired by and what is the major difference, e.g., sub-trajectory balance vs. transition-based learning (assume that reader is also not familiar with the sub-trajectory balance).
>
> We do not use the sub-trajectory balance objective in this work. Sub-trajectory balance is to take sub-trajectory (could be longer than a single transition but also could be shorter than a complete trajectory) to construct training losses. On the other hand, transition-based learning is only applicable to detailed balance based methods (including detailed balance and forward-looking detailed balance). It has the same expected gradient as trajectory-level detailed balance training, but just use the formula in line 238 rather than Eq. 5 to make the optimization more efficient. We will include the discussion into the final version.
>
> > Finally, there are several neural combinatorial optimizations works [1,2] that study the symmetric nature of combinatorial optimization. I humbly suggest to include as a reference paper.
> [1] Kim, Minsu, Junyoung Park, and Jinkyoo Park. "Sym-nco: Leveraging symmetricity for neural combinatorial optimization." Advances in Neural Information Processing Systems 35 (2022): 1936-1949.
> [2] Kim, Hyeonah, et al. "Symmetric Exploration in Combinatorial Optimization is Free!." arXiv preprint arXiv:2306.01276 (2023).
>
> Thanks for the references! We will make sure to include them in the final version.
>
> > How does diversity (e.g., the edit distance between sampled solution) changes when your new techniques are applied? (to compare with another method, including the trajectory balance method)?
>
> In the MIS small scale benchmark, we test the diversity of different GFlowNet variants. For each graph configuration in the test set, we let each algorithm generate thirty solutions and compute their average pairwise distance. The diversity is the mean distance averaged across the test set. The distance is computed as the Jaccard distance between two solution vertex sets. The diversity of trajectory balance, detailed balance (trajectory level), detailed balance (transition level), forward-looking detailed balance (trajectory level), and forward-looking detailed balance (transition level) is $0.714$, $0.572$, $0.637$, $0.505$, and $0.618$, respectively.
>
> > What kinds of post-processing methods are used for solution generation? Did you samples multiple samples from the learned sampler? Then what is the actual sampling width? How about applying local search just as learning what to defer (LwD)?
>
> Yes, we sample multiple times from the learned sampler and do not use any further post-processing technique; see Appendix C for details. We also agree that applying local search with GFlowNet solutions will definitely be a promising future direction for this.
>
> > To compare with LwD, did you put 2opt for LwD following their original paper?
>
> LwD does not provide code for local search; also in the LwD paper, only part of the experiments are done with 2opt. To ensure a fair comparison, we also do not put 2opt after GFlowNet and compare it with other baselines including LwD without 2opt.
>
> > Can this method extend to other combinatorial optimization, such as the traveling salesman problem (TSP)?
>
> TSP problems are one of the future directions where we hope to see the advantage of GFlowNets on this family of combinatorial optimization problems. For example, the action of GFlowNet could be to choose the next vertex to visit (for the traveling salesman). The architecture of policy and flow could be a transformer like in previous approaches [1].
>
> [1] Attention, Learn to Solve Routing Problems!

---

> > ### Comment · Reviewer_PEdA · 2023-08-10
> >
> > Thank you for your timely response. My concerns are resolved. Also, I find the extension of GFlowNet into TSP through the utilization of the AM [1] particularly intriguing. I uphold a positive evaluation of this work, as it delivers a pivotal initial contribution to the realm of combinatorial optimization within the framework of GFlowNet.
> >
> > [1] Attention, Learn to Solve Routing Problems!

---

> > > ### Author Response · Authors · 2023-08-14
> > > **Thank you to Reviewer PEdA!**
> > >
> > > Thank you for the kind reply.
> > >
> > > We appreciate the reviewer's recognition of the novelty and promising performance of our approach. We would also like to thank the reviewer for the time and effort in reviewing our work and considering our rebuttal, which has greatly contributed to the improvement of our paper!

---

### Official Review · Reviewer_VHLn · 2023-07-04

**Soundness:** 3 good
**Presentation:** 3 good
**Contribution:** 2 fair
**Rating:** 5
**Confidence:** 3

**Summary:**

This paper presents a new approach for learning to solve graph combinatorial optimization by GFlowNets. The GFlowNets (or broadly, the generative models) own the advantage of discovering multiple (near)-optimal solutions, which is better than standard RL or SL. The authors modify GFlowNets to work with larger-scale problems in CO and provide extensive experiments on 4 different problems.

---------------------

Post-rebuttal: I appreciate the authors for the feedback and I am now more positive about this paper.

**Strengths:**

* This paper is well-written.
* The motivation for introducing GFlowNets to CO is sound and interesting.
* The authors made modifications to GFlowNetw to fit larger-scale CO problems.

**Weaknesses:**

* My major concern of this paper is about the selection of baselines and the implementations in experiments.
  * The authors mentioned in L55 that "there are attempts to fix these issues (Kwon et al., 2020; Ahn et al., 2020)" from the RL literature, but these methods are not implemented and compared in experiments.
  * Why the selection of baselines are different for MIS and the other 3 problems, especially considering that MIS and Max clique are complement?
  * What will the performance be like if we let Gurobi run the same amount of time compared to your GFlowNet solvers?
  * During inference, do you implement a search algorithm after the GFlowNets? What about the other neural solver baselines?
  * How do you implement, configure and tune the baselines? Do you implement the same GIN as your implementation of GFlowNets? For example, from my knowledge, PPO's performance is strongly dependent on hyperparameter tuning and the configuration of tricks. Besides, the probabilistic method (Karalias & Loukas, 2020) has two types of configurations (fast/accurate) and which one do you choose?
  * I also notice you follow some recent neural CO solvers such as Qiu et al. (2022, DIMES) and Sun & Yang (2023, DIFUSCO) but do not consider them in main experiments. Can you offer any justifications?
* The training process does not seem clear to me. Is the training objective to minimize $l_{DB}+l_{TB}$? How is the reward being used? Seeing that GFlowNet is quite a new framework, we should not expect the readers to be that familiar with the domain knowledge.
* Some details seem to be missing:
  * What is the meaning of "MIS size" in Fig 4? On which dataset did you get the numbers in Fig 4?
* Other minor issues:
  * The notation of $P_F^\top$ is improper. As I understand, $\top$ does not mean "transpose" here and causes confusion to readers.

**Questions:**

Please see the questions in the "weaknesses" part.

---

> ### Author Rebuttal · Authors · 2023-08-09
>
> > The authors mentioned that "there are attempts to fix these issues (Kwon et al; Ahn et al.)" but these are not compared in experiments.
>
> Please note that we have compared with Ahn et al, which is the “PPO” in Table 1. The Kwon et al. work is designed for routing problems (e.g., TSP)  which are different from the problems we study in this work, thus we don't include it. That said, we will definitely take routing problems as one of the future directions where we hope to see the advantage of GFlowNets on this family of combinatorial optimization problems.
>
> > different baselines between MIS and others
>
> Our choice of baselines was guided by a desire to provide a comprehensive comparison across a diverse set of methods. For the MIS problem, we have already included a substantial number of baselines in our benchmark. Notice that most baselines are specialized for MIS and not applicable for other tasks. For the other problems, we follow the benchmark that is used by [1], aiming to include a broader range of baselines, particularly those from the unsupervised learning domain.
>
> Regarding the complementarity of the MIS and Max Clique problems, we agree with your point. To cover a diverse set of baselines on max clique, we add an experiment on the twitter dataset following Sec. 4 of [2] and [3]. In this benchmark, the methods of [2], [3], [4], and GFlowNet achieve average results of $0.926$, $0.924$, $0.987$, and $0.992$. This demonstrates the excellence of GFlowNet performance. Our goal was to cover a diverse set of baselines within our limited time. By comparing these, we provide a more comprehensive evaluation.
>
> [1] Annealed Training for Combinatorial Optimization on Graphs
> [2] Unsupervised Learning for Combinatorial Optimization with Principled Objective Relaxation
> [3] Erdos Goes Neural:an Unsupervised Learning Framework for Combinatorial Optimization on Graphs
> [4] Graph Neural Networks for Maximum Constraint Satisfaction
>
> > run Gurobi in limited time
>
> We do experiments on running Gurobi on large scale MIS task with limited time budget. We give a time budget that is slightly larger than GFlowNet to Gurobi, and it obtains the average size of $34.81$, smaller than $37.48$ of GFlowNet. Therefore, Gurobi obtains worse performance compared to GFlowNets under the same time budget constraint.
>
> > a search algorithm during inference?
>
> We do not implement any search algorithm based on GFlowNets (see the evaluation part of Appendix C). That being said, it will be a very promising future direction to explore to further boost the performance of GFlowNet solvers. The Intel and DGL baselines are built on a tree search algorithm, while other baselines do not use search algorithms. Thus it's fair to say that GFlowNet achieves better performance.
>
> > How do you implement, configure and tune the baselines?
>
> For different methods, we use the same evaluation and hyperparameter selection protocols to make sure the comparison is fair (see Appendix D). For RL baseline we stick to configurations in [1] to ensure that we are not “creating” a worse baseline that is different from the original previous work.
>
> [1] Learning What to Defer for Maximum Independent Sets
>
> > probabilistic method has two configurations... which one do you choose?
>
> We implement the conditional sequential decoding proposed in Erdos [1], which is slower but more accurate than directly doing monte-carlo sampling (see Sec. 3.3 in [1]). Therefore, it is fair to say that our comparison is reasonable.
>
> [1] Erdos Goes Neural: an Unsupervised Learning Framework for Combinatorial Optimization on Graphs
>
> > ... some recent neural CO solvers but do not include them
>
> That isn't true. We do include their methods in Appendix D to show the advantage of GFlowNets especially on large scale tasks. In the main text we don't include them as they don't use the same graph simulation protocol as ours and we also got bugs when trying to reproduce their results on our data. During the week after main text submission deadline, we did experiments on ER graphs with GFlowNets and thus could compare with them, so we put these results into Appendix D. The results show the advantage of GFlowNet. Further, our approach is a purely unsupervised method while theirs rely on precomputed solutions by the solver. Thus it is not fair to compare unsupervised methods with supervised methods. On the other hand, this means GFlowNet uses less information but still achieves better results.
>
> > Is objective $\ell_{TB} + \ell_{DB}$? How is the reward used?
>
> No; in fact, the forward-looking (FL) objective in Eq. 6 is a modification of the detailed balance objective with (one can see the similarity between Eq. 2 and Eq. 6) that uses cumulative reward information in the parametrization of the state flow. For DB-based methods, the reward is used as the state flow of the last state in a complete trajectory. We will make sure to specify the usage of reward information in the final version of the paper.
>
> > What is "MIS size" in Fig 4? On which dataset to get Fig 4?
>
> "MIS size" is the size of the vertex set that an algorithm obtains for a maximum independent set task. The larger the size, the better the method. More details about evaluation metrics are written in the paragraph starting from line 313. As specified in line 338, we conduct experiments on a small scale simulated RB graph dataset to produce Fig. 4.
>
> > The notation of $P_F^{\top}$ is improper. As I understand, $\top$ does not mean "transpose" here and causes confusion to readers.
>
> Here $\top$ is for “terminating” and $P_F^{\top}$ is the distribution of terminating states induced by GFlowNet’s forward policy $P_F$ (line 88). This notation is borrowed from past work on GFlowNets and we will replace the improper notation.

---

> > ### Comment · Reviewer_VHLn · 2023-08-13
> >
> > Thanks to the authors for the feedback. I believe the quality of this paper will be greatly improved if the authors clarify the aforementioned details and include new results in future revisions. I adjust the score to 5.

---

> > > ### Author Response · Authors · 2023-08-14
> > > **Thank you to Reviewer VHLn!**
> > >
> > > Thank you for increasing the score.
> > >
> > > We appreciate the reviewer's recognition of the sound and interesting motivation behind our paper, as well as our careful and novel design of the approach, and the acknowledgment of the quality of the writing.
> > >
> > > We would also like to thank the reviewer for the time and effort in reviewing our work and considering our rebuttal, which has greatly contributed to the improvement of our paper!

---

### Official Review · Reviewer_gYzR · 2023-07-05

**Soundness:** 3 good
**Presentation:** 4 excellent
**Contribution:** 3 good
**Rating:** 7
**Confidence:** 4

**Summary:**

This paper leverages generative flow networks (GFlowNets) to obtain diverse solution candidates for combinatorial optimization problems on graphs without expert supervision.

GFlowNets were recently introduced as a way to sample structured objects $\mathbf{x}$ with a likelihood $P(\mathbf{x}) \propto R(\mathbf{x})$ that is proportional to a terminal reward function (see _Flow Network based Generative Models for Non-Iterative Diverse Candidate Generation_, Bengio et al. 2021).
This is done by casting the sampling as a Markov Decision Process, whose policy is defined by neural networks and trained with a suitable loss to enforce flow balance conditions.

In the study at hand, the reward is chosen as a Gibbs probability distribution $R(\mathbf{x}) = \exp(\mathcal{E}(\mathbf{x}) / T)$, whose temperature controls solution diversity by specifying the acceptable deviation from the global optimum.
Parametric policies are constructed conditionally on the graph, to allow for generalization.
MDPs are designed in a problem-specific manner, by iteratively constructing sets of vertices that satisfy certain constraints.
Since the MDP trajectories can be quite long, two tricks are suggested to improve training efficiency and credit assignment:
- sample random subsets of transitions instead of a full trajectory
- incorporate intermediate reward signals based on partial solutions

These strategies are implemented and benchmarked against exact solvers, as well as recent ML-based approaches.

**Strengths:**

### Originality

This paper mixes several existing ideas into a coherent whole:

- GFlowNets and their loss functions
- viewing optimization as probabilistic inference
- MDPs for combinatorial problems
- strategies for credit assignment

This unseen combination, along with extensive numerical experiments, is sufficient novelty in my view.

### Quality

The authors demonstrate deep knowledge of the literature on GFlowNets, and the theoretical part is well supported.
Thorough benchmarks suggest their method performs better than its counterparts, although some comparisons may be biased (see below).
I especially appreciated the care given to make the benchmarks transparent and fair (aligning with recent papers, avoiding easy instances, reimplementing and retraining algorithms from scratch to compare them on the same basis).
Additionally, a very welcome ablation study shows the limited impact of hyperparameter settings.
In a nutshell, this is good, serious science.

### Clarity

The writing style is clear and nicely complemented by explanatory pictures.
A lot of time is devoted to the prerequisites on GFlowNets and combinatorial problems, as well as a thorough literature review.

### Significance

I believe this contribution can inspire further developments, since the use of GFlowNets for combinatorial optimization seems well-justified and practically successful.

**Weaknesses:**

### Originality

To me, it appears that the main novel idea in the paper is the application of GFlowNets to learning combinatorial problems (which is already a worthy one).
The idea of training from subsampled transitions instead of a full trajectory is also new.

However, I would be very surprised if the MDPs presented by the authors were absent from the previous literature, especially given the standard nature of problems such as Maximum Independent Set.
Perhaps there are some necessary adjustments that are necessary for these MDPs to work with GFlowNets, in which case I will welcome clarification!

### Quality

I am not confident enough to comment on the choice of algorithms in the MIS benchmark, but I have an issue with the 3 non-MIS benchmarks, which seem a bit unfair to me.
A major asset of GFlowNets is their sequential construction of the solution, which the authors mention as the reason for their success.
On the other hand, the benchmarked competitors ERDOS and ANNEAL rely on a hypothesis of independence between vertices of the graph, which explains their fast inference.
In light of this, I argue that it would make sense to benchmark GFlowNets against other methods (typically from the RL literature) that also adopt a sequential perspective, to see if the advantage still holds.

### Clarity

n.a.

### Significance

To underline the impact of the paper, it would be helpful to explain why a diverse set of solution candidates is important in practice.
As a combinatorial optimizer myself, it is not obvious to me: when should I settle for a set of diverse, but possibly suboptimal solutions?

**Questions:**

L61: What is new in your MDP design compared to existing formulations?

L105: Where do the final rewards appear?

L242: I thought the whole point was to make epochs faster?

L255: Are there cases where this reward continuation is not applicable?

L640: Why not use a linear formulation? What formulation did you use for the three non-MIS problems?

**Limitations:**

The authors do not really discuss the limitations of their approach, and more reflection on that would be welcome.

Societal impact is not relevant in this case.

---

> ### Author Rebuttal · Authors · 2023-08-09
>
> > To me, it appears that the main novel idea in the paper is the application of GFlowNets to learning combinatorial problems (which is already a worthy one). The idea of training from subsampled transitions instead of a full trajectory is also new. … However, I would be very surprised if the MDPs presented by the authors were absent from the previous literature, especially given the standard nature of problems such as MIS. Perhaps there are some necessary adjustments that are necessary for these MDPs to work with GFlowNets, in which case I will welcome clarification!
>
> > L61: What is new in your MDP design compared to existing formulations?
>
> We do not aim to claim that we “invent” the MDPs, but only want to find efficient algorithms to solve these MDPs. We acknowledge that the design of our MDPs, which is simple yet effective, is very straightforward. In fact, [1] uses similar MDPs as ours; they design MDPs that “add one vertex at each step”, but on minimum vertex covering, maximum cut, and traveling salesman problem. That said, they do this on slightly different CO problems, and they also use different reward designs from ours. Other MDP designs include [2] where each action is to “edit” the current solution, and [3] where each action is to add multiple vertices into the solution set. We will include this discussion to the final version.
>
> [1] Learning Combinatorial Optimization Algorithms over Graphs
> [2] Learning to Perform Local Rewriting for Combinatorial Optimization
> [3] Learning What to Defer for Maximum Independent Sets
>
> > I am not confident enough to comment on the choice of algorithms in the MIS benchmark, but I have an issue with the 3 non-MIS benchmarks, which seem a bit unfair to me. A major asset of GFlowNets is their sequential construction of the solution, which the authors mention as the reason for their success. On the other hand, the benchmarked competitors ERDOS and ANNEAL rely on a hypothesis of independence between vertices of the graph, which explains their fast inference. In light of this, I argue that it would make sense to benchmark GFlowNets against other methods (typically from the RL literature) that also adopt a sequential perspective, to see if the advantage still holds.
>
> A large part of research in combinatorial optimization is about trading-off the performance and efficiency. One of the advantages of Erdos-based methods including Anneal is the efficiency; on the other hand, even given a large enough time budget (like what we do in the experiments), it will not obtain tremendous improvement. Please note that we have compared with the PPO baseline (from a recent related work [4]) in MIS comparison, which is close to GFlowNet in the sense of sequential generation (but fails to encourage diversity in the solutions -- please find our detailed explanation as below), and we will incorporate more comparison of RL baselines with sequential generation behavior in the final version.
>
> [4] Learning What to Defer for Maximum Independent Sets
>
> > To underline the impact of the paper, it would be helpful to explain why a diverse set of solution candidates is important in practice. As a combinatorial optimizer myself, it is not obvious to me: when should I settle for a set of diverse, but possibly suboptimal solutions?
>
> As discussed in the paragraph starting at line 44, there could be several cases where we want to emphasize diversity. For example, there could be multiple different optimal solutions in combinatorial optimization problems due to the symmetry of problem configurations, as pointed out in [1]. In addition, diversity in the solutions holds great importance across several dimensions, while solely looking for an optimal solution for the current problem could fail:: **Robustness** Multiple solutions can provide a form of robustness. If one solution fails or is not feasible due to changes in the problem environment or constraints, having alternative solutions can be very useful; **Exploration of the Solution Space** A diverse set of solutions allows for a broader exploration of the solution space. This can provide more insight into the structure of the problem and could serve as warm starts for numerical solvers; **Stakeholder Preferences** In many real-world problems, there may be multiple stakeholders with different preferences or objectives. A diverse set of solutions can provide a range of options that cater to these different preferences; **Dynamic Environments** In dynamic environments where the problem parameters can change over time, having a diverse set of solutions can allow for quick adaptation and better generalization to these changes.
>
> [1] Combinatorial optimization with graph convolutional networks and guided tree search
>
> > L105: Where do the final rewards appear?
>
> Thanks for the question. When a state is terminal, then its flow value would be the reward value, which is the source of learning signal grounded by the environment.
>
> > L242: I thought the whole point was to make epochs faster?
>
> The point is to show that our proposed novel GFlowNet variant is better within the same training time. Since different variants consume similar wall clock time for one epoch, we could roughly think of epoch as a measure of training time.
>
> > L255: Are there cases where this reward continuation is not applicable?
>
> To the best of our knowledge, this has demonstrated applicability across the problems we have studied, particularly those where all states belong to the same space.
>
> > L640: Why not use a linear formulation? What formulation did you use for the three non-MIS problems?
>
> We follow what is provided in the MIS benchmark [1]. We thus try two different Gurobi formulations, which achieve average size of $40.14$ in $2:15:07$,  and average size of $40.90$ in $2:10:36$, respectively. For other problems we use linear formulations.
>
> [1] https://github.com/MaxiBoether/mis-benchmark-framework

---

> > ### Comment · Reviewer_gYzR · 2023-08-14
> >
> > > We do not aim to claim that we “invent” the MDPs, but only want to find efficient algorithms to solve these MDPs.
> >
> > As early as the abstract (L8) and later in the intro (L61), you state "we design MDPs for different combinatorial problems". It is the verb "design" that I find misleading in this case. Perhaps "adapt" or "leverage" would be better suited.
> >
> > > We will include this discussion to the final version.
> >
> > Good idea.
> >
> > > Please note that we have compared with the PPO baseline (from a recent related work [4]) in MIS comparison, which is close to GFlowNet in the sense of sequential generation (but fails to encourage diversity in the solutions -- please find our detailed explanation as below), and we will incorporate more comparison of RL baselines with sequential generation behavior in the final version.
> >
> > Perfect, thank you.
> >
> > > In addition, diversity in the solutions holds great importance across several dimensions, while solely looking for an optimal solution for the current problem could fail [...]
> >
> > Great explanation, I think it would really strengthen the paper to include it early on, reworking the paragraph starting at L44 in the process.
> >
> > > The point is to show that our proposed novel GFlowNet variant is better within the same training time. Since different variants consume similar wall clock time for one epoch, we could roughly think of epoch as a measure of training time.
> >
> > Perhaps I misinterpreted what you mean by "epoch" here. In the case where you subsample the transitions by a factor of $k$, an epoch involves $k$ times more passes? In that case, indeed, the plot is very convincing!
> >
> > > We follow what is provided in the MIS benchmark [1]
> >
> > I don't think that's true. I followed the link you gave and ended up on this paper describing the benchmark: https://arxiv.org/pdf/2201.10494.pdf. In Section 2, they state that they use a linear formulation for MIS in the main paper. This is justified in Appendix D2, where they indeed _try_ the quadratic one but conclude that the linear option performs better in nearly all cases.

---

> > > ### Author Response · Authors · 2023-08-15
> > >
> > > Thanks for your suggestions and we will improve the writing in revision accordingly.
> > >
> > > > I followed the link you gave and ended up on this paper describing the benchmark: https://arxiv.org/pdf/2201.10494.pdf. In Section 2, they state that they use a linear formulation for MIS in the main paper. This is justified in Appendix D2, where they indeed try the quadratic one but conclude that the linear option performs better in nearly all cases.
> > >
> > > Notice that we quote the GitHub link instead of their paper. In their GitHub implementation, both formulations are provided. In the early stage of this project, we found that the linear formulation sometimes got error with some weird hardware error information that we fail to identity; thus we run with the quadratic formulation instead. In the rebuttal period, we try two different formulations of Gurobi in MIS large scale tasks, and they got average size of $40.14$ (linear formulation) and $40.90$ (quadratic formulation) respectively. This indicates that the difference of formulation is not significant in our data setting.

---

> > > > ### Comment · Reviewer_gYzR · 2023-08-18
> > > >
> > > > Alright, thank you for clarifying!

---

### Official Review · Reviewer_n6CJ · 2023-07-06

**Soundness:** 3 good
**Presentation:** 4 excellent
**Contribution:** 3 good
**Rating:** 6
**Confidence:** 4

**Summary:**

The authors propose an approach for using GFlowNets for solving some NP-Hard combinatorial optimization problems on graphs. The GFlowNet approach is designed to better perform credit assignment for fitting a constructive policy that adds nodes on the graph to a currently considered subset. The authors propose slight modifications of GFlowNet that help it perform well on graph optimization tasks which require long trajectories and improved credit assignment. The authors justify their design decisions empirically via ablation studies. Finally, the authors evaluate the proposed approach against supervised learning, unsupervised learning, heuristics, and traditional OR approaches where applicable. They evaluate on a variety of problems considering max independent set, max clique, minimum dominating set, and max cut. They demonstrate improved solution quality over the investigated baselines and mostly improved runtime over OR baselines. Overall, the paper is a promising initial attempt at applying GFlowNets to solve CO problems on graphs.

**Strengths:**

The main strength of the paper is in addressing limitations of previous RL for CO approaches using GFlowNet. In previous RL-based optimization approaches, the issue of credit assignment is a major challenge as often individual choices have limited immediate impact on the overall solution quality while the long-term impact can be quite large. GFlowNet as an approach aims to perform this credit assignment well as it aims to learn a policy based on a long-term reward.

A second strength is that the authors evaluate the given approach against a variety of baselines on a number of settings and present results on all settings.

Lastly, the writing is quite clear, and the paper gives a useful introduction of how to use GFlowNet for CO problems which can likely lead to future work of employing GFlowNet for broader types of CO problems.


**Weaknesses:**

The main weakness is that the main contribution of the paper is that it is mainly directly applying GFlowNet to CO on graphs with minor modifications of the GFlowNet approach, and justifying their approach with intuition and empirical results. It might be good to either identify 1) specific ways that the GFlowNet methodology might be adapted to be specially suited to CO problems, or 2) ways in which the particularities of GFlowNet might be especially suited for answering interesting questions in CO problems.

Some ideas for 1) it may be helpful to formulate the architecture used in GFlowNet to encode certain properties that help solvers such as being invariant to permutation of the solution generation procedure, special actions for GFlowNet such as deleting components of the current solution. Other ideas might be to modify GFlowNet to make use of alternative information such as supervision from known solutions on the training dataset, precomputed primal solutions with objective values, or integrating solver components like presolve etc.

For 2) it might be helpful to showcase the ability to generate diverse solutions by evaluating the diversity of solutions generated by the GFlowNet in terms of how many unique solutions it produces. Alternatively, one could generate a solution pool and using that to warm start solvers or generate solutions for multi-objective optimization problems.

Overall, it would be interesting to see how the GFlowNet approach could be used outside of improving solve time.


While the empirical results are promising, it is somewhat unfair to compare the given heuristic methods only against gurobi as an exact solver, since gurobi may be spending a lot of time proving optimality in addition to finding the optimal solution. It would be more reasonable to compare against gurobi which is tuned for quickly finding primal solutions and which is set at a very strict time limit, since gurobi may be finding the optimal solution quickly and taking most of its time proving optimality. Additionally, gurobi seems to be solving faster than the proposed approach in the MDS setting.


**Questions:**

[Paragraph starting at line 44] One listed downside of RL in CO is that RL may not produce diverse solutions. It would be helpful to demonstrate how this approach addresses that for instance by demonstrating that the GFlowNet approach does generate diverse solutions compared to the RL approach. For instance, Gurobi is able to return a pool of solutions. It would be interesting to compare the solution diversity between approaches.

For the forward looking technique, would it be possible to use the objective value of a “completed solution” which finds the optimal solution given the partial solution as fixed? Does this need to be a differentiable function of the solution?

It would be helpful to give the variance for the results to determine how close the distribution of results are since they are somewhat close to previous work. Additionally, it would be helpful to give some metrics such as win rate, i.e. the percent of instances for which the given approach was fastest / got the best quality solution.


It would also be helpful to explain the sizes of the train / test datasets and how they were split especially in the case of SATLIB.

It is unclear how this approach might handle CO problems where feasibility is “difficult”. For instance, for TSP on non-complete graphs it may be hard to find a feasible tour. It would be helpful to explain how this method might approach settings where feasibility is nontrivial via constructive methods. Here it may be difficult to sample feasible solutions in the first place and as such it may be helpful to understand how this method would perform. Similarly, it would be helpful to explain how this method would perform in settings where the decisions are more complex than selecting a subset of vertices. This is somewhat hinted at in the paragraph starting on line 195 but it would be helpful to explain this limitation and how it might be addressed.


Small questions:
[line 29] Why is gurobi considered as giving approximate solutions? It not only should give optimal solutions given enough runtime but should also give a bound on the solution quality in order to prove optimality.

[line 258] handcraft -> handcrafted
Table 1, Time of Ours in Large has a leading 0 before the 4:22


**Limitations:**

Relevant limitations are addressed

---

> ### Author Rebuttal · Authors · 2023-08-09
>
> > directly apply GFlowNet to CO with minor modifications
>
> See the general response.
>
> > formulate architecture to encode certain properties… modify GFlowNet to make use of alternative information
>
> We appreciate the reviewer's insightful suggestions about interesting future directions. Indeed, we use a standard architecture and don't exploit the geometric properties, and the standard scenarios as in previous GFlowNets works that don't allow deleting actions. We acknowledge that these are valuable future directions which could further boost the GFlowNet performance. This indicates that there is still huge potential of GFlowNet in CO problems. Furthermore, our approach is purely unsupervised (without the knowledge of precomputed solutions). On the other hand, this can be an interesting future direction: e.g., we can start from a given solution, and then use the GFlowNet's backward policy to generate trajectories which we know would lead to informative high-reward regions. These high-quality trajectories could be very useful in training as they save a lot of exploration effort.
>
> > diversity of GFlowNet and other methods
>
> We compute the diversity of RL and GFlowNet. For each graph in testset, we let each algorithm generate 30 solutions and compute their mean pairwise distance. The diversity is the mean distance across testset. The distance is computed as the Jaccard distance between two solution vertex sets. In the MIS small task RL and GFlowNet got $0.428$ and $0.618$, respectively. As for Gurobi, it obtains a diversity of $0.312$ when being used with pool solution feature to give multiple solutions. This indicates that GFlowNet produces more diverse samples than PPO; notice that GFlowNet also achieves better performance than PPO on this benchmark in the sense of a larger solution set.
>
> > unfair to compare given methods only against Gurobi… Gurobi faster than GFlowNet on MDS on large task
>
> Indeed, the configuration of using Gurobi to solve combinatorial optimization problems could be flexible and using different formulations might cause different performance. However, we do involve KaMIS, which is a highly specialized solver for MIS problems, in our experiments; we believe KaMIS will achieve at least similar performance to the best tuned Gurobi if not better. What’s more, our Gurobi results are from [1], which has been adopted by many other CO related machine learning papers. Furthermore, our method does use heuristics in algorithmic design, thus it would be unfair to be compared with Gurobi with specialized heuristics. Lastly, we try two different formulations of Gurobi in MIS large scale tasks, and they got average size of $40.14$ in $2:15:07$,  and average size of $40.90$ in $2:10:36$, respectively.
>
> While it's true that our method didn't outperform Gurobi on MDS with a large scale dataset, it's important to consider the overall performance. Due to Occam's razor, different methods have different strengths and weaknesses, and their performance vary depending on the characteristics of dataset. The dataset in question may have certain unique characteristics that particularly favor the baseline method. Our method demonstrated superior or competitive performance on almost all datasets, which we believe is a strong indication of its effectiveness and robustness.
>
> [1] https://github.com/MaxiBoether/mis-benchmark-framework
>
> > For forward looking, would it be possible to use... optimal solution? Does this need to be differentiable?
>
> It would be a great and feasible idea to use known optimal solutions. For example, in MIS if we know the optimal solution is of size $a$ and the current set has $b$ nodes, then we can use $b/a\in [0, 1]$ to serve as the negative partial energy $\mathcal{\tilde E}$. We will need to accordingly use $c/a\in [0, 1]$ as the terminating reward where $c$ is the solution set size of the terminating state. In this example, we do not require differentiability.
>
> > variance for the results… other metrics
>
> We repeat 5 runs of GFlowNet, which achieve $19.20\pm 0.08$ on the MIS small scale task and $37.89\pm 0.36$ on the large scale task. This indicates that the performance of GFlowNet is stable and the improvement upon other baselines is statistically significant. Due to limited time, we only calculate the win rate of GFlowNet against Gurobi, which is $0.326$. Notice that this is a nontrivial result as Gurobi is almost perfect on this benchmark, which indicates the advantage of GFlowNets.
>
> > train/test split
>
> As described in Appendix D, for tasks with RB/BA graph we use a training set of size 4000, a validation set of size 500, and a test set of size 500; for SATLIB we use a training set of size 39000, a validation set of size 500, and a test set of size 500. For RB and BA graphs we do uniformly splitting as the data are simulated in the same way for train/valid/test; for SATLIB, we obtained the split from the authors of [1].
>
> [1] DIFUSCO:Graph-based Diffusion Solvers for Combinatorial Optimization
>
> > when feasibility is “difficult”
>
> Indeed, we ensure all obtained solutions are feasible through specific MDP design. For more general CO problems it could be hard to get a feasible solution, let alone to achieve a solution that maximizes some target property. That being said, this poses a significant challenge not only to our method but to all ML approaches in general. To address this issue, one potential approach could be to use a very negative reward when getting infeasible results. This could encourage the model to avoid infeasible solutions. Alternatively, we could consider a soft penalty approach, where the reward is a monotonic function of the number of constraints satisfied. This would allow the model to learn to improve feasibility over time when we gradually use a temperature to make the soft constraint harder.
>
> > Why is Gurobi considered as approximate?
>
> You are right. We originally wanted to say “give approximate solutions with limited time” and will correct this in revision.

---

> > ### Comment · Reviewer_n6CJ · 2023-08-16
> > **followup**
> >
> > I thank the authors for their response and clarifications which have addressed my concerns. I believe this is a good initial work towards solving graph CO problems with gflownet that opens several new followup directions for future work in using gflownet for CO.

---

> > > ### Author Response · Authors · 2023-08-16
> > >
> > > We would like to sincerely thank the reviewer who recognized the value of our work and contributed to raising the score of the submission. Our work benefits a lot through the rebuttal. Your expertise, effort, and attention to detail truly made a difference, and we are sincerely thankful for your dedication to the peer-review process.

---

### Author Rebuttal · Authors · 2023-08-09

## General Response

We sincerely thank all the reviewers for your insightful comments and suggestions.

> About the contribution and novelty of our work

While our approach builds upon the general GFlowNet framework, we have made significant designs and innovations to ensure its scalability and effectiveness in tackling complex CO problems which cannot be achieved with direct application. First, our MDP formulations are specially designed for different CO problems and have never been studied in previous GFlowNet works. These designs are crucial to ensure the solution satisfying the feasibility requirement (this is related to another question raised by the same reviewer related to “difficult” feasibility CO). Second, the GFlowNet training scale in this work is much larger in the sense of GPU occupation than any previous ones, and we would like to emphasize that our work pushes the boundaries of GFlowNet's training scale. To cope with this challenge, we develop novel training techniques to enable efficient learning (Sec. 3.3) which serve as a remedy to issues about training efficiency and memory consumption for previous approaches [1]. These techniques are not only applicable in CO tasks but can also be extended to any GFlowNet applications such as molecule design. On the other hand, there are a large number of papers [2] on using RL to solve CO problems and people do not consider them as “direct application”, let alone for GFlowNets which we have discussed to be more appropriate than RL for these tasks.

[1] Trajectory balance: Improved credit assignment in GFlowNets
[2] Reinforcement Learning for Combinatorial Optimization: A Survey

---

### Decision · Program_Chairs · 2023-09-21

**Decision:**

Accept (spotlight)

**Comment:**

The paper has obtained reviewer consensus in favor of acceptance and the AC follows it. The scores and positive reception merit spotlight. Overall the paper is a good combination of GFlowNets and a number of other augmentations with nice results. The rebuttals have been noted as well as their positive influence on the review scores.